# Study on the Causes of Differences in Cropland Abandonment Levels among Farming Households Based on Hierarchical Linear Model—13,120 Farming Households in 26 Provinces of China as an Example

**Xiangdong Wang** [1,2] and **Decheng Zhao** [1,*]

1 School of Management, Lanzhou University, Lanzhou 730000, China; wangxiangdong@lzu.edu.cn
2 Institute of County Economic Development & Rural Revitalization Strategy, Lanzhou 730000, China
* Correspondence: zhaodch21@lzu.edu.cn

**Abstract:** Cropland abandonment is a widespread phenomenon with an increasing trend around the world, including China. Cropland abandonment is the result of a multi-scale and multifactor nested structure. In order to comprehensively identify the individual and background effects, this study explores the causes of the differences in farmland abandonment levels at two levels: farm household and village, based on a hierarchical linear model. The results show that (1) 83.63% of the differences in farm abandonment rates are differences in farm households, while 16.37% are differences in their villages; (2) several factors, including whether the head of household is healthier, per capita cropland area, ratio of transferred farmland, possession of large-scale agricultural production machinery or livestock for agricultural production, ratio of agricultural income, and whether there are village cadres, have a significant negative impact on abandonment rates at the farm household level, while at the village level, commuting distance, whether it is a suburb of a large or medium-sized city, topography of the village is plain or not, and the ratio of the number of people in agricultural production in the village have a significant positive impact on abandonment rates. Furthermore, whether it has experienced land expropriation has a significant positive effect on the abandonment rate; (3) commuting distance weakens the negative correlation between the ratio of transferred cropland and the ratio of agricultural income and the abandonment rate. In addition, whether it is a suburb of a large or medium-sized city strengthens the negative correlation between whether it owns large-scale agricultural production machinery or livestock and the abandonment rate, and whether the topography of the village is plain weakens the negative correlation between the possession of large agricultural production machinery or livestock for agricultural production and the agricultural income ratio and the abandonment rate. Furthermore, ratio of the number of people in agricultural production in the village weakens the negative correlation between the ratio of transferred cropland and abandonment rate, and whether the village has experienced land expropriation strengthens the negative correlation between the ratio of agricultural income and abandonment rate.

**Keywords:** cropland abandonment; hierarchical linear model; background effect; causal analysis; China



## 1. Introduction

Cropland abandonment is an important manifestation of land use change [1], characterized by a long existence and wide distribution. In recent years, by the global new crown epidemic rampant, changes in the international trade situation of agricultural products and the frequent occurrence of extreme disasters and weather effects, cropland abandonment in the world is becoming more and more intense [2]. Since the 1980s, the phenomenon of farmland abandonment has been highlighted in China, which has been aggravated with the deepening of urbanization and industrialization and the continuous withdrawal

of agricultural laborers [3]. According to statistics, in 2011 and 2013, 13.5% and 15% of China's agricultural land was idle, respectively [4]. Farmland abandonment is a common problem faced by the whole world, which has become an important direction of land use/cover research (LUCC) [3], and the study and solution of this problem are conducive to maintaining regional and global food security and realizing the sustainable development goal of eradicating hunger, which is even more significant for China, a populous country. Therefore, the problem of cropland abandonment should be given special attention in China.

In response to the problem of abandonment, scholars at home and abroad have conducted rich discussions on the connotation, causes, impacts, and countermeasures of abandonment from the perspectives of agronomy, ecology, geography, and agroforestry economics [5]. Among them, research on the causes can be outlined from three aspects, namely influencing factors, research scale, and analysis methods. First, regarding influencing factors, the abandonment of cropland is formed under the influence of a variety of factors, such as economic development, agricultural policies, labor force characteristics, location conditions, and natural conditions [6–9], with the influence of socio-economic factors predominating [3,9]. By generalization, these factors can be divided into three categories, namely socio-economic conditions, labor force characteristics, and agricultural production conditions [3], and each category of factors can be further divided into specific categories, for example, socio-economic conditions include production costs and incomes [10–12], policies and institutions [13,14], and other socio-economic factors [15,16]; labor force characteristics cover labor force quantity [17,18], labor force quality [19,20], and other labor force factors [21,22]; and agricultural production conditions can be divided into natural production conditions [23,24] and social production conditions [25,26]. Second, regarding the research scale, including the observation scale and interpretation scale [27], the so-called observation scale is the scope of research, and the interpretation scale is the analysis perspective. In terms of the observation scale, there are differences at home and abroad; domestic scholars mostly explore the problem of abandonment in small areas such as counties, townships, and villages [28,29], while foreign scholars focus on the larger scope of the country and region [14,30]. In terms of the interpretation scale, both domestic and foreign scholars favor a relatively microscopic perspective, such as villages, farmers, and plots of land [31–33], and focus on the simultaneous use of multiple scales of interpretation. Third, regarding the analysis method, the existing abandonment research methods can be divided into qualitative research methods and quantitative research methods, of which the qualitative methods include field survey method, literature analysis method, and qualitative comparative analysis (QCA) [34–37], while the quantitative methods contain the logit model, tobit model, probit model, OLS model, regression tree model, and hierarchical linear model (hierarchical linear model), according to the different methods of abandonment level measurement. Of these methods, including the logit model, tobit model, probit model, OLS model, regression tree model, and hierarchical linear model (HLM), etc. [15,26,38,39], the traditional regression analysis method is relatively frequently used.

In summary, through the identification of influencing factors, the scale of explanation and the determination of analytical methods, the existing research has explored the causes of abandonment in multiple regions and continuously confirmed that the abandonment of cropland is the result of the integrated effect of multi-scale and multifactor analysis, but there are still deficiencies in the following two aspects: first, the analysis method is not properly selected, which leads to insufficient scientific and accurate conclusions about the causes. Influenced by factors at different levels, cropland abandonment has a multilevel data structure, which is the result of the nested structure of factors at various levels [39]. However, the traditional regression analysis methods used in existing studies combine data at different scales and analyze the causes of abandonment purely based on the individual level, ignoring the influence of background effects on the individual [40,41]. The so-called background effects refer to the data with a nested structure in social science research and the role of the relationship between the variables across the hierarchical levels [41,42], so

that many differences brought about by subgroups are interpreted as individual differences, ultimately resulting in flawed conclusions. The hierarchical linear model can scientifically deal with multilevel structured data and is currently more successfully applied in the fields of education, psychology, health, and organizational management [43], and some scholars have also used this model to explore the causes of abandonment, but all of them suffer from the problem of inadequate explanation of background effects, which is reflected in the failure to scientifically analyze the interactions between the factors of each level, especially the influence of high-level factors on the slope of low-level factors. Second, the scope of the existing cause analysis in China is mostly small, and the conclusions obtained are limited in representativeness and generality, so there is an urgent need to carry out research on the causes of cropland abandonment in a more macro-regional context.

Based on the results and shortcomings of the existing case studies, this study explores the causes of cropland abandonment in 26 provinces in China based on the CLDS data in 2014, 2016, and 2018, using a hierarchical linear model to identify individual and background effects at the household and village levels, with the aim of obtaining more scientific and general conclusions about the causes of cropland abandonment and promoting better solutions to the problem of cropland abandonment.

## 2. Theoretical Analysis

### 2.1. Push and Pull Mechanisms for the Emergence of Abandonment Behavior

Farmers are the main body of cropland use, as well as the "rational man", in pursuit of their own interests as the goal [44,45], driven by the reduction in the benefits of agricultural production and the attraction of increased benefits of non-agricultural employment, choosing cropland abandonment.

On the one hand, due to the promotion of urbanization and industrialization, changes in market demand, fluctuations in agricultural prices, adjustments in agricultural-related policies and systems, and advances in agricultural technology, the price of labor, seeds, fertilizers, pesticides, etc., rises the price of agricultural products falls, and the phenomenon of the marginalization of farmland becomes more and more serious, and it has become an important driving force for farmers to choose to leave farmland abandoned. The so-called marginalization of cropland refers to when a piece of cropland has only one alternative use due to changes in production costs and incomes that make cropland's profits fall to zero or negative; at this time, cropland is in the no-rent margin outside, and no matter how the farmers adjust the input of factors of production, cropland is in the no-rent margin outside. Consequently, as a "rational person" the farmers will choose not to operate this piece of arable land, resulting in the abandonment of the farmland [3].

On the other hand, the process of urbanization and industrialization will create a large number of non-farm employment opportunities, which provides the possibility for farmers to maximize their interests and exerts a pulling force on farmers to choose to abandon their farmland. With a large influx of the rural population into the city, the livelihood strategy of farm households has gradually changed to part-time employment and non-farming, and the maximized benefits they can obtain can be subdivided into two aspects: economic gains and non-economic gains [46]. From the perspective of economic gain, generally speaking, the part-time and non-farming livelihood strategy has higher economic gain compared with traditional agriculture, and can maximize economic gain; from the perspective of non-economic gain, due to the long-term existence of the urban–rural dichotomy, there is a big difference between the city and the countryside in terms of social security system, medical conditions, educational resources, etc. [47]. The agricultural households moving from the countryside to the city can also enjoy the more superior living conditions in the city and maximize non-economic gains.

### 2.2. Internal and External Constraints on the Choice of Abandonment Behavior

Cropland abandonment, as one of the livelihood strategies of farm households, has the motive of maximizing benefits behind its choice. However, from the perspective of the

sustainable livelihood analysis framework, the adjustment of farmers' livelihood strategies is not arbitrary but is constrained by their own financial, physical, natural, human, and social livelihood capital [48]. At the same time, the theory of motivated behavior points out that, in addition to the influence of internal livelihood capital, the role of external natural and socio-economic factors should not be underestimated [48,49], and it is necessary to avoid the problem of ignoring the external environmental context or confusing it with livelihood capital [50]. The physical geography and socio-economic characteristics of villages have the closest and most direct relationship with farm households, and the cropland abandonment behaviors of different farm households within the same village are generally more similar than those of randomly selected farm households in different villages. This shows that when choosing whether or not to abandon cropland, farmers have to make a comprehensive consideration of their own livelihood capital and village characteristics.

In addition, village characteristics and farmers' livelihood capital are constraints from different levels and have a relatively complex nested structure in their impacts on farmers' cropland abandonment behaviors, so the two cannot be simply merged and conflated, but rather, we should fully consider the individual and background effects on the process of village- and farmer-level factors and pay attention to the independent impacts of the factors at each level, as well as analyze the interactions among the factors at different levels, especially focusing on the role of the village-level factors in terms of the impacts of farmer-level factors on cropland abandonment.

### 2.3. Theoretical Analysis Framework

The above analysis clearly demonstrates why farmers choose to leave their farmland abandoned, as well as the internal and external constraints they face when choosing to leave their cropland abandoned, which is used as the basis for constructing a theoretical analytical framework with a focus on the multilevel structure of village- and household-level factors. The theoretical analysis framework is shown in Figure 1.

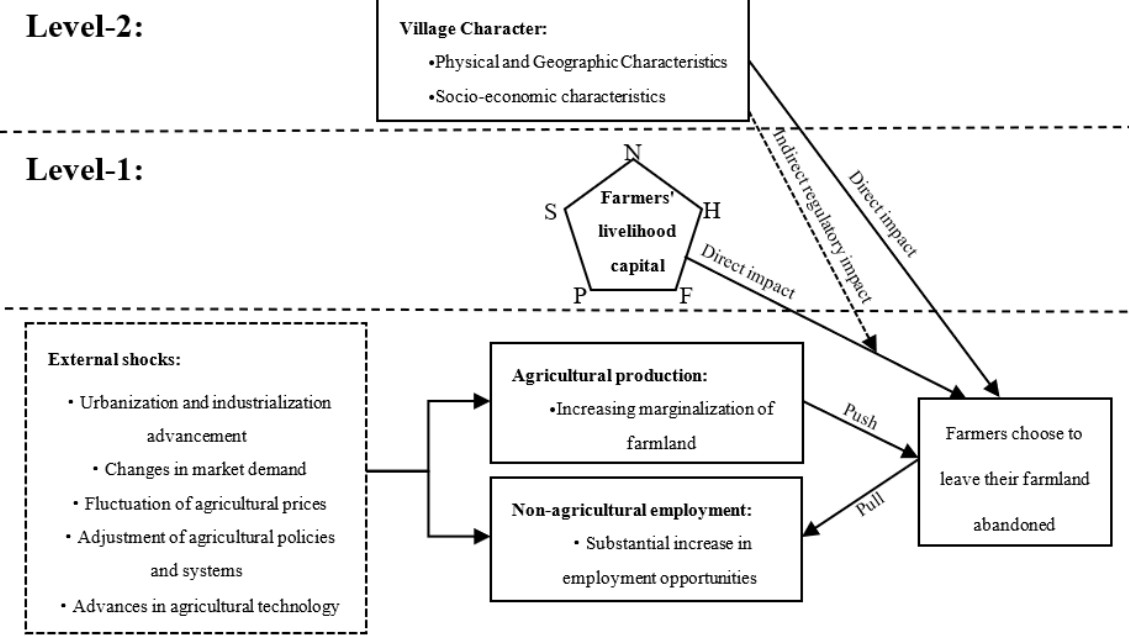

**Figure 1.** Theoretical analysis framework.

The analytical framework shows that, under the influence of external forces such as urbanization and industrialization, the marginalization of farmland for agricultural

production has been worsening, while the opportunities for non-farm employment have been increasing. Additionally, it shows that farmers, as "rational beings", faced with the dual influence of the push in the field of agricultural production and the pull in the field of non-farm employment, will shift their livelihood strategy from traditional agriculture to part-time work and non-farming, resulting in the abandonment of farmland. However, the process of farmers adjusting their livelihood strategies to continue maximizing their benefits is not arbitrary but is subject to a variety of internal and external factors, mainly manifested in farmers' livelihood capital and village characteristics. It is worth noting that whether farmers choose to leave their farmland abandoned or not is the result of the nested structure of factors at the farmer and village levels, and it is important to fully identify the individual and background effects and to explore the moderating role of village-level factors on the effects of farmer-level factors on farmland abandonment based on the clarification of the effects of factors at each level on the abandonment of farmland.

## 3. Research Methodology and Data Sources

### 3.1. Research Methodology

In this study, a hierarchical linear model was chosen to investigate the causes of the differences in the level of cropland abandonment, specifically covering four sub-models: the null model, the random-effects regression model, the intercept model, and the complete model. The principle of the hierarchical linear model to deal with nested structured data is as follows: the regression equation is first established with the explained variables and level-1 explanatory variables, after which the intercept and slope in the equation are used as the explained variables, and a secondary regression is conducted using the level-2 explanatory variables; this treatment can decompose the variance of the explained variables into the two parts of the within-group and between-group variance [41], and by decomposing the variance, the model is able to distinguish between individual and background effects and reveal the relationship between individual and organizational variables. The basic form of the model is as follows:

Level-1 model:

$$QGZB_{ij} = \beta_{0j} + \sum_{p=1}^{n} \beta_{pj} X_{pij} + \gamma_{ij} \tag{1}$$

Level-2 model:

$$\beta_{pj} = \gamma_{p0} + \sum_{q=1}^{m} \gamma_{pq} W_{qj} + \mu_{pj}$$

In Equation (1), $i$ represents the farm household; $j$ represents the village in which the farm household is located; $n$ represents the number of level-1 explanatory variables and $p$ is the number of its values; $m$ represents the number of level-2 explanatory variables and $q$ is the number of its values. $QGZB_{ij}$ is the abandonment rate; $\beta_{0j}$ is the intercept of the regression of the level-1 explanatory variables on the explanatory variables, i.e., the average abandonment rate of farm households in the $j_{th}$ village; $X_{pij}$ denotes the $p_{th}$ explanatory variable of level-1 and $\beta_{pj}$ is its corresponding regression coefficient; $\gamma_{ij}$ is the stochastic component of level-1; $\gamma_{p0}$ denotes the intercept of the regression of level-2 explanatory variables on $\beta_{pj}$, i.e., the overall average of the abandonment rate of all the villages; $\gamma_{pq}$ denotes the slope of the regression of level-2 explanatory variables on $\beta_{pj}$; $W_{qj}$ denotes the $q_{th}$ explanatory variable of level-2; and $\mu_{pj}$ is the random component of level-2.

### 3.2. Variable Identification and Indicator System

#### 3.2.1. Explained Variables

To measure cropland abandonment, there are two types of variables commonly used in academia, namely discrete and continuous, of which the discrete variable is whether or not to abandon the cropland, and the continuous variable is the area of abandonment and

the rate of abandonment of cropland. In this study, we believe that the use of the dummy variable of whether to abandon or to not abandon reflects the abandonment behavior of farm households, which will lose a lot of important information [15]; at the same time, there is a significant difference in the size of the cultivated area in each region, and the abandonment area ignores the positive correlation between the number of abandoned areas and the total area of cropland, which is also insufficient for this measurement. In light of this, in order to more accurately measure the phenomenon of cropland abandonment, the abandonment rate, i.e., the ratio of the area of cropland left abandoned by households to the total area of cropland in the household, was selected as the explanatory variable in this study.

3.2.2. Explanatory Variables

(1)   Farm-household-level (level-1) explanatory variables

Based on the livelihood capital in the sustainable livelihood analysis framework, this study selects farm-household-level explanatory variables from the five aspects of human, natural, physical, financial, and social capital, taking into account existing research and data availability. ① Human capital: covering the quantity and quality of labor, this study selects the size of the household to characterize the quantity of labor; the head of the household, as the head of the family, has an important influence on the choice of household livelihood strategies, and selects whether the head of the household has a high school education or higher and whether the head of the household is physically healthier to characterize the quality of labor. ② Natural capital: the focus is on tangible productive capital, mainly cropland resources, which is the objective object of abandonment. This study analyzes the quantity and quality of cropland, of which the per capita area of cropland and the ratio of transferred cropland characterize the quantity of cropland, and whether there is pollution of cropland characterizes the quality of cropland. ③ Physical capital: this includes infrastructure and public services and the means of production. In China, the government sector, as the main provider of infrastructure and public services, strongly guarantees that farmers can enjoy basically the same infrastructure and public services in each region. Therefore, this study focuses on the means of production of farm households, which are characterized by whether they own large-scale agricultural production machinery used for agricultural production or whether the number of livestock and household durable goods reaches five or more types. ④ Financial capital: this mainly examines the inflow of funds in order to more accurately and comprehensively reflect the inflow of funds to the farm household per capita in this study, and the proportion of the two dimensions, the total annual income of the family per capita, agricultural income and non-agricultural income of the three aspects of the logarithm of the total annual income of the family per capita, and the share of agricultural income and non-agricultural income accounting for the proportion of the three indicators were selected. ⑤ Social capital: this is mainly derived from social organizations and social networks. In this study, participation in social organizations is characterized by whether or not there is a village in the ministry, and the logarithm of annual human expenditure and whether or not they use the Internet are selected to characterize social networks.

(2)   Village-level (level-2) explanatory variables

In this study, village-level explanatory variables were selected from the aspects of natural geographic characteristics and socio-economic characteristics. ① Physical geographic characteristics: these mainly reflect the topography and location of the village, covering three indicators, namely commuting distance, whether it is a suburb of a large or medium-sized city, and the topography of the village. ② Socio-economic characteristics: these emphasize the industrial development and policy implementation of the village, including the ratio of the number of agricultural producers in the village, the presence of non-agricultural enterprises, whether the implementation of agricultural services, and whether the village has experienced land expropriation.

### 3.2.3. Indicator System

Based on the results of variable identification, a system of indicators of the causes of abandonment was constructed. At the same time, the direction of the variables was predicted by combining the existing case studies, the "+" indicates that the factor has a positive effect on the abandonment of cropland, the "−"indicates that the factor has a negative effect on the abandonment of cropland, and the "±"indicates that the factor may have a positive or negative effect on the abandonment of cropland.

The system of indicators is shown in Table 1.

**Table 1.** Indicator system of causes of cropland abandonment.

| Level | Variable Type | Variable Group | Variable Name and Symbol | Variable Definition and Assignment | Intended Effect |
|---|---|---|---|---|---|
| Farm household level | Explained variable | | Abandonment rate (QGZB) | Proportion of land abandoned by farmers to total cultivated area (%) | |
| | Explanatory variable | Human capital | Do household heads have a high school diploma or higher (EDU) | No = 0; Yes = 1 | + |
| | | | Whether the head of household is in relatively good health (HEAL) | No = 0; Yes = 1 | − |
| | | Natural capital | Household size (JTRKGM) | Sum of persons living with the family and persons away from the family (person) | − |
| | | | Cropland area per capita (RJGDMJ) | Ratio of area of cropland owned by households to total population size (acre/person) | − |
| | | | Ratio of transferred cropland (ZRGDB) | Proportion of area of cropland transferred by households to total area of cropland (%) | − |
| | | Physical capital | Presence of contaminated cropland (TRWR) | No = 0; Yes = 1 | − |
| | | | Ownership of large agricultural production machinery or livestock for agricultural production (SCGJ) | No = 0; Yes = 1 | − |
| | | | Whether the number of types of durable goods in the household is five or more (NYZL) | No = 0; Yes = 1 | + |
| | | Financial capital | Total log annual household income per capita (LNRJSR) | Logarithm of the ratio of the sum of all types of household income in a year to the size of the household (yuan) | + |
| | | | Agricultural income ratio (NYSRB) | Share of annual household income from agriculture in total annual household income (%) | − |
| | | | Non-farm income ratio (FNSRB) | Household annual non-farm income as a share of total annual household income (%) | + |
| | | Social capital | Availability of village cadres (CGB) | No = 0; Yes = 1 | − |
| | | | Log of annual expenditure on favors (LNRQZC) | Total household expenditure on gifts and gratuities in a year (yuan) | ± |
| | | | Use of the Internet or not (HLWQK) | No = 0; Yes = 1 | ± |

**Table 1.** *Cont.*

| Level | Variable Type | Variable Group | Variable Name and Symbol | Variable Definition and Assignment | Intended Effect |
|-------|---------------|----------------|--------------------------|-----------------------------------|-----------------|
| Village level | Explanatory variable | Physical geographic characteristics | Commute distance (TQJL) | Distance of villages from township offices (kilometers) | − |
| | | | Whether it is a suburb of a medium-sized city (DLWZ) | No = 0; Yes = 1 | + |
| | | | Topography of the village is plain or not (CZDX) | No = 0; Yes = 1 | − |
| | | Socio-economic characteristics | Number of village agricultural producers (CZNYSCB) | Percentage of people engaged in agriculture in villages compared to the total number of people in villages (%) | − |
| | | | Availability of non-farm enterprises (FNQY) | No = 0; Yes = 1 | + |
| | | | Whether or not to implement agricultural services (HNFW) | No = 0; Yes = 1 | − |
| | | | Experiencing land expropriation or not (TDZY) | No = 0; Yes = 1 | + |

*3.3. Data Sources and Processing*

3.3.1. Data Source

The data for this study come from the China Labor Dynamics Survey Database (CLDS), which is constructed under the auspices of the National Development Research Institute of Sun Yat-sen University and currently has four issues of data published for the years 2012, 2014, 2016, and 2018, covering the three levels of the individual, the household, and the village. The samples cover 29 provinces and municipalities in China, which are of a certain degree of scientific validity, authoritativeness, comprehensiveness, and national representativeness [51].

3.3.2. Data Processing

According to the purpose of the study, this study selected three periods of data in 2014, 2016, and 2018, mainly using the household and village residence questionnaires, and based on the screening of the required variables, the three years of data were combined into unbalanced panel data and processed with the help of Stata16.0 and SPSS22.0 software, totaling 40,751 samples. The specific processing process includes the following: (1) retaining the samples that live in rural areas and the head of household is an agricultural household while contracted with cropland; (2) deleting the samples that exist in the cases of inapplicability, lack of clarity, and refusal to answer, etc.; (3) deleting the samples that do not conform to the significance of economics, which are mainly the samples that have negative incomes in various categories; (4) deleting the samples that have the head number of the household missing and decomposing the remaining samples into the two datasets of farm households and villages; (5) dealing with the missing values in the dataset as the maximum missing proportion of both datasets is less than 10%, and the mean interpolation method is used to interpolate the missing values, in which the continuous indicators are interpolated using the mean of the series, and the discrete indicators are interpolated using the plurality of numbers [52]; (6) shrinking the tail of the continuous indicators in the two datasets, to exclude the impact of outliers on the regression results; (7) generating the proportion indicators required for the study and excluding the samples that do not have economic significance, such as the proportion greater than 1, etc. At the same time, logarithmic treatment is applied to the income and expenditure indicators to improve the accuracy of the regression results; (8) re-matching the datasets of farm households and villages based on the village number to form the final dataset, covering 13,120 farm households, 645 villages, and 26 provinces, the data of which are nationally representative. The extent of the study area and the distribution of the samples are shown in Figure 2.

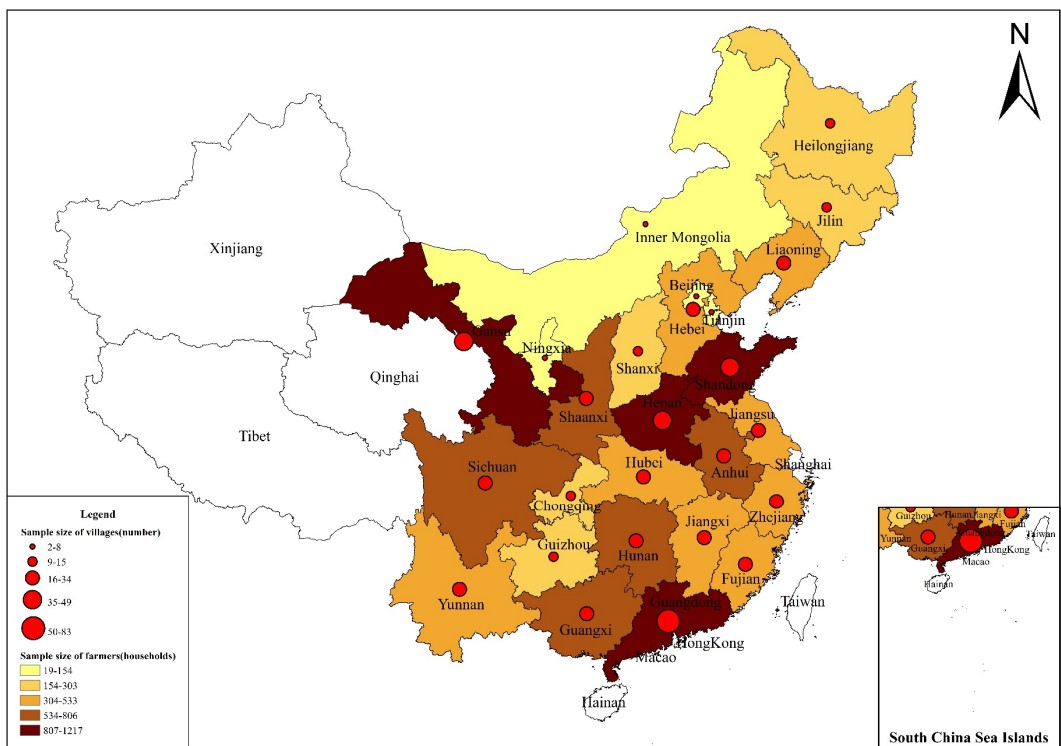

**Figure 2.** Extent of the study area and sample distribution.

### 3.3.3. Descriptive Statistics of Variables

Descriptive statistics of the selected explained and explanatory variables were performed using HLM 6.08 software. The results of the descriptive statistics are shown in Table 2. Meanwhile, the average household abandonment rate for each province in the study area is presented in Figure 3.

**Table 2.** Results of descriptive statistics of variables.

| Variable Type | Variable Name | Sample Size | Mean | Standard Deviation | Min | Max |
|---|---|---|---|---|---|---|
| Explained variable | QGZB | 13,120 | 8.75 | 24.92 | 0.00 | 100.00 |
| | EDU | 13,120 | 0.09 | 0.29 | 0.00 | 1.00 |
| | HEAL | 13,120 | 0.54 | 0.50 | 0.00 | 1.00 |
| | JTRKGM | 13,120 | 5.17 | 2.75 | 1.00 | 15.00 |
| | RJGDMJ | 13,120 | 1.68 | 2.58 | 0.02 | 50.00 |
| | ZRGDB | 13,120 | 5.58 | 18.65 | 0.00 | 100.00 |
| | TRWR | 13,120 | 0.49 | 0.50 | 0.00 | 1.00 |
| Farm-household-level explanatory variables | SCGJ | 13,120 | 0.12 | 0.32 | 0.00 | 1.00 |
| | NYZL | 13,120 | 0.39 | 0.49 | 0.00 | 1.00 |
| | LNRJSR | 13,120 | 8.39 | 1.21 | 0.18 | 12.21 |
| | NYSRB | 13,120 | 36.85 | 42.70 | 0.00 | 100.00 |
| | FNSRB | 13,120 | 39.77 | 44.09 | 0.00 | 100.00 |
| | CGB | 13,120 | 0.01 | 0.10 | 0.00 | 1.00 |
| | LNRQZC | 13,120 | 5.39 | 3.65 | 0.00 | 9.90 |
| | HLWQK | 13,120 | 0.41 | 0.49 | 0.00 | 1.00 |
| | TQJL | 645 | 5.68 | 5.22 | 0.00 | 30.00 |
| | DLWZ | 645 | 0.07 | 0.26 | 0.00 | 1.00 |
| | CZDX | 645 | 0.41 | 0.49 | 0.00 | 1.00 |
| Village-level explanatory variables | CZNYSCB | 645 | 64.51 | 34.09 | 0.00 | 100.00 |
| | FNQY | 645 | 0.27 | 0.45 | 0.00 | 1.00 |
| | HNFW | 645 | 0.84 | 0.37 | 0.00 | 1.00 |
| | TDZY | 645 | 0.47 | 0.50 | 0.00 | 1.00 |

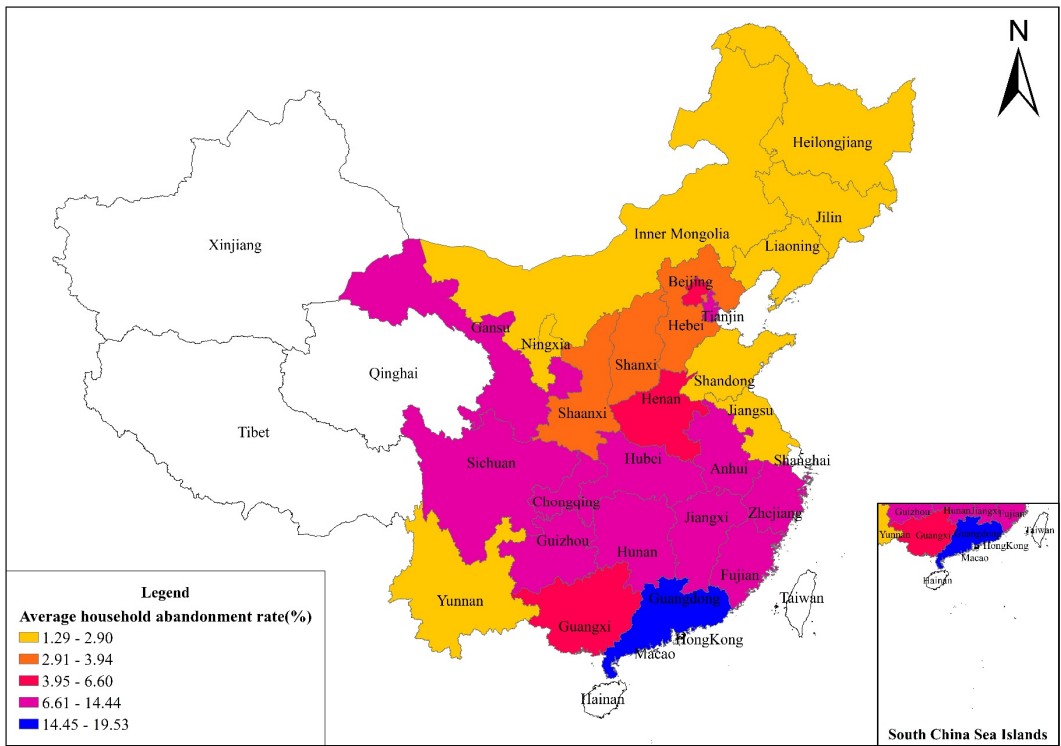

**Figure 3.** Average household abandonment rate in the study area.

## 4. Results and Analysis

### 4.1. Normality Test of the Explanatory Variables

The hierarchical linear model requires the explained variables to satisfy the condition of obeying normal distribution, which is loosely approximated to satisfy the condition of obeying normal distribution [53]. Therefore, in this study, the skewness and kurtosis tests were chosen to test the normality of the abandonment rate with the help of SPSS 22.0 software.

Table 3 shows a skewness value of 2.953 (standard error 0.021), Z-score = 2.953/0.021 = 140.619 and a kurtosis value of 7.420 (standard error 0.043), Z-score = 7.420/0.043 = 172.558. The skewness and kurtosis values are both >0, and the Z-score is not in the range of ±1.96, which indicates that the that the explanatory variables do not obey normal distribution. However, some studies in academia have shown that when the kurtosis value is 7.420 less than 10 and the skewness value is 2.953 less than 3, this indicates that the data are not absolutely normally distributed but basically acceptable as normal distribution [54]; therefore, the data are capable of stratification study.

**Table 3.** Results of normality test for explained variables.

| Variable Name | Sample Size | Mean | Standard Deviation | Skewness | | Kurtosis | |
|---|---|---|---|---|---|---|---|
| | | | | Skewness Value | Standard Error | Kurtosis Value | Standard Error |
| QGZB | 13,120 | 8.751 | 24.924 | 2.953 | 0.021 | 7.420 | 0.043 |

### 4.2. Null Model Analysis

The so-called null model, which is a model with no explanatory variables in each level of the model, is the first step in the analysis of the hierarchical model, aiming to decompose the overall variance of the abandonment rate at the levels of farmers and villages and to judge the necessity of the level-2 model. The specific form of the model is shown below:

Level-1 model:

$$QGZB_{ij} = \beta_{0j} + \gamma_{ij} \tag{2}$$

Level-2 model:

$$\beta_{0j} = \gamma_{00} + \mu_{0j}$$

In Equation (2), the variables have the same meaning as in Equation (1).

The results of the null model regression are shown in Table 4. The results of model confidence estimation showed that the confidence of the level-1 intercept was estimated to be 0.765, and the higher the confidence, the smaller the variance of the error, and the closer the estimated farm abandonment rate fitted by the model was to the actual abandonment rate. In general, if the model estimated confidence > 0.5, the model can be considered to basically meet the requirements [55]. Fixed effects showed that the overall mean of the abandonment rate for all villages was 9.0206%. The chi-squared test for random effects showed that there was an extremely significant difference in the rate of farm abandonment at the village level ($p$ = 0.000), i.e., the village-level factors had an important influence on the rate of farm abandonment, and it was necessary to add some explanatory variables in the level-2 model. Using the intragroup correlation coefficient (ICC) formula, the intragroup correlation coefficient of the model can be calculated as 101.9664/(101.9664 + 521.0150) = 0.1637. Generally speaking, this value is greater than 0.059, which belongs to the medium degree of intragroup correlation, and it can be applied to the HLM analysis [56,57]. This value indicates that 16.37% of the variance in farm abandonment rate is village variance and the remaining 83.63% is farm household variance. This shows that although farm household differences are the main cause of differences in abandonment rates, the effect of village differences on them should not be underestimated.

**Table 4.** Null model estimation results.

| Fixed Effects and Significance Test | | | | Random Effects and Significance Test | | | | |
|---|---|---|---|---|---|---|---|---|
| Parameters | Regression Coefficient | $t$-Test | $p$-Value | Parameters | Standard Deviation | Variance Component | Chi-Squared Test | $p$-Value |
| $\gamma_{00}$ | 9.0206 | 19.8610 | 0.000 | $\mu_0$ | 10.0978 | 101.9664 | 3164.5207 | 0.000 |
| | | | | $r$ | 22.8258 | 521.0150 | | |

### 4.3. Random-Effects Regression Model Analysis

The random-effects regression model, which included only level-1 variables and did not include village-level variables in the equation, was used to determine the effect of level-1 variables on the explanatory variables and to determine whether the intercept and regression coefficients of level-1 variables had significant variation at the village level. In order to improve the accuracy of the full model, this study first used a one-factor random-effects regression model to investigate the influence of individual explanatory variables of level-1 on the abandonment rate [57]; after that, we screened the factors with significant influence in the one-factor model and constructed a multifactor random-effects regression model.

#### 4.3.1. One-Factor Random-Effects Regression Model Analysis

Taking the abandonment rate as the explained variable, level-1 variables were introduced one by one in a non-centralized way to construct a one-factor random-effects regression model, and no variables were introduced in the level-2 model. The effect of farmer-level variables on abandonment rate was tested by the fixed-effects statistics of the model. The specific form of the model is shown below:

Level-1 model:

$$QGZB_{ij} = \beta_{0j} + \beta_{1j}[EDU_{ij}/HEAL_{ij}/JTRKGM_{ij}/RJGDMJ_{ij}/ZRGDB_{ij}/TRWR_{ij}/SCGJ_{ij} \\ /NYZL_{ij}/LNRJSR_{ij}/NYSRB_{ij}/FNSRB_{ij}/CGB_{ij}/LNRQZC_{ij}/HLWQK_{ij}] + \gamma_{ij} \tag{3}$$

Level-2 model:

$$\beta_{0j} = \gamma_{00} + \mu_{0j}$$
$$\beta_{1j} = \gamma_{10} + \mu_{1j}$$

In Equation (3), $EDU_{ij}$ is the education level of the head of the household of the $i$th farm household in the $j$th village, and the rest of the level-1 explanatory variables have similar meanings and represent the livelihood capital situation of each farm household in each village; the other variables have the same meanings as in Equation (1).

The fixed effects results of the one-factor random effects regression model are shown in Table 5, indicating that the human capital, natural capital, physical capital, financial capital, and social capital of farmers have different degrees of influence on the abandonment rate.

**Table 5.** Fixed-effects estimates of one-factor random-effects regression models.

| Variable Name | Regression Coefficient | Standard Error | *t*-Test | *p*-Value |
|---|---|---|---|---|
| EDU | −0.5046 | 0.6864 | −0.735 | 0.462 |
| HEAL | −1.6655 | 0.4355 | −3.824 | 0.000 |
| JTRKGM | −0.0124 | 0.1004 | −0.124 | 0.902 |
| RJGDMJ | −0.5925 | 0.0523 | −11.341 | 0.000 |
| ZRGDB | −0.0617 | 0.0086 | −7.213 | 0.000 |
| TRWR | −0.4463 | 0.5051 | −0.884 | 0.378 |
| SCGJ | −4.3581 | 0.5950 | −7.325 | 0.000 |
| NYZL | −0.6327 | 0.4236 | −1.494 | 0.136 |
| LNRJSR | −0.3407 | 0.2185 | −1.559 | 0.119 |
| NYSRB | −0.0872 | 0.0061 | −14.313 | 0.000 |
| FNSRB | 0.0431 | 0.0061 | 7.078 | 0.000 |
| CGB | −2.7299 | 1.2966 | −2.105 | 0.035 |
| LNRQZC | −0.0250 | 0.0641 | −0.390 | 0.697 |
| HLWQK | −0.2000 | 0.4305 | −0.465 | 0.642 |

(1) In human capital, whether the head of household is healthier or not has an extremely significant negative effect on the rate of the abandonment of farming, indicating that the healthier the head of household is, the lower the rate of abandonment of farming. The head of the household plays a major role in determining the direction and content of the family's production and management activities, and the healthier they are, the more energy and stamina they will have to engage in a variety of production activities, including agricultural production, and the degree of abandonment of cropland will be less [49,58]. (2) In natural capital, both the per capita area of cropland and the ratio of transferred cropland have extremely significant negative effects on the abandonment rate of farm households, indicating that the larger the per capita area of cropland and the higher the ratio of transferred cropland, the lower the abandonment rate of farm households. The amount of cropland is an important safeguard for cropland production; the larger the per capita area of cropland, the better the cropland resource endowment of the farmers, and the more convenient the scale and mechanization of production and operation, the more the cropland yield and agricultural income are guaranteed, and consequently, the cropland is not easy to be left abandoned [58,59]. The purpose of the farmers transferring to the farmland of other people is the same in order to realize the scale of operation and to seek a higher agricultural income, so when there is a larger area in the total area of their cropland transferring to the cropland, in order to obtain a cost that is greater than the cost of the chartered land in terms of farm income, the farmers are inclined to continue to engage in agricultural production [60]. (3) In physical capital, the possession of large agricultural production machinery or livestock for agricultural production has an extremely significant negative effect on the abandonment rate of farm households, which

suggests that the abandonment rate is lower for farmers who own agricultural production machinery or livestock. Agricultural production machinery or livestock as supporting facilities for agricultural production can make farmers operate a larger area of cropland, improve yields, increase agricultural income, enough to reduce the input of agricultural labor, prompting farmers to have the opportunity to engage in a variety of production and management activities; consequently, the rate of abandonment of cropland is not easy to be abandoned [61,62]. (4) In financial capital, the agricultural income ratio has an extremely significant negative effect on the abandonment rate of farm households, while the non-farm income ratio has an extremely significant positive effect on the abandonment rate of farm households, which indicates that the higher the agricultural income ratio, the lower the abandonment rate of farm households [63,64], and the higher the non-farm income ratio, the higher the abandonment rate of farm households. The agricultural income ratio and the non-farm income ratio reflect the differences in the main sources of income for farm households, and farm households that rely on agricultural income are less likely to abandon their farmland, whereas regarding those who are predominantly non-agricultural income earners, their farmland is more prone to abandonment [20,65]. (5) In social capital, the presence or absence of village cadres has a significant negative effect on the abandonment rate of farm households, which suggests that farm households with village cadres have a lower abandonment rate. Farmers with village cadres in their households are more likely to understand the policy requirements for farmland protection, have a higher degree of awareness of the serious situation and importance of farmland protection, and need to play a good role as a demonstration of farmland protection, and therefore are more inclined to engage in agricultural production [49,66].

### 4.3.2. Multifactor Random-Effects Regression Model Analysis

Based on the statistical results of the one-factor model, a multifactor random-effects regression model is constructed. Taking the abandonment rate as the explained variable, the level-1 variables in the one-factor model, which have a significant effect on the abandonment rate, are added to the level-1 model in a non-centralized way, and no variables are added to the level-2 model. The specific form of the model is shown below:

Level-1 model:

$$QGZB_{ij} = \beta_{0j} + \beta_{1j}HEAL_{ij} + \beta_{2j}RJGDMJ_{ij} + \beta_{3j}ZRGDB_{ij} + \beta_{4j}SCGJ_{ij} + \beta_{5j}NYSRB_{ij} \\ + \beta_{6j}FNSRB_{ij} + \beta_{7j}CGB_{ij} + \gamma_{ij} \tag{4}$$

Level-2 model:

$$\beta_{0j} = \gamma_{00} + \mu_{0j}$$
$$\beta_{1j} = \gamma_{10} + \mu_{1j}$$
$$\beta_{2j} = \gamma_{20} + \mu_{2j}$$
$$\beta_{3j} = \gamma_{30} + \mu_{3j}$$
$$\beta_{4j} = \gamma_{40} + \mu_{4j}$$
$$\beta_{5j} = \gamma_{50} + \mu_{5j}$$
$$\beta_{6j} = \gamma_{60} + \mu_{6j}$$
$$\beta_{7j} = \gamma_{70} + \mu_{7j}$$

In Equation (4), the variables have the same meaning as in Equations (1) and (3).

Comparing the fixed-effects estimation results of single-factor and multifactor random-effects regression models, it can be found that the overall impact trend in regression coefficients of most of the indicators remains the same, and there are only different degrees of slight changes in coefficient sizes, which may be due to the existence of multiple covariates among multiple variables in the model [57]; among them, the impact of the non-farm income ratio on the rate of abandonment of farmland is changed from a significant positive effect to a non-significant effect, and multiple covariation of the indicator may be relatively serious, which affects the regression results [57]; therefore, the present study considers elim-

inating the non-farm income ratio in the complete model in order to increase the reliability of the model's regression results. Meanwhile, the fixed-effects estimation results show that the overall average of the abandonment rate of all villages is 13.2053%; the abandonment rate will decrease by 1.9220% if the head of the household is healthier; the abandonment rate will decrease by 0.1843% for every increase in per capita cropland area; the abandonment rate will decrease by 0.0374% for every increase in the ratio of transferred cropland by 1%; and for every increase in the ratio of transferred cropland by 1% and every increase in the ratio of households owning large agricultural production for agricultural purposes, a further decrease is present. The abandonment rate decreases by 2.3390% for households owning large agricultural machinery or livestock used for agricultural production; the abandonment rate decreases by 0.0792% for every 1% increase in the ratio of agricultural income; and the abandonment rate decreases by 3.8157% for households owning village cadres.

The results of the random-effects estimates are used to characterize whether there is significant two-level variation in the regression coefficients of the intercept term and level-1 variables. As can be seen from Table 6, (1) the chi-squared test result of the intercept term is significant, with a *p*-value of 0.000, indicating that there are significant village differences in the mean value of farm abandonment rate among villages, and it is necessary to add the village-level variables in the subsequent analysis for the analysis of background effects; (2) the chi-squared test results of the regression coefficients of the ratio of the transferred to cropland, the possession of large-scale agricultural production machinery or livestock used for agricultural production, and the ratio of the farm income on the abandonment rate of the farmers are significant, with *p*-values of 0.085, 0.040, and 0.000, respectively, that is, among different villages, there are significant inter-group differences in the negative impacts of the ratio of transferred cropland, the possession of large-scale agricultural production machinery or livestock used for agricultural production, and the ratio of agricultural income on the rate of abandonment of farming by farm households, and it is necessary to add village-level explanatory variables in the full model to clarify the moderating effect of their effects on the impacts of household-level variables on the rate of abandonment; (3) the negative effects of other variables are relatively consistent across villages.

**Table 6.** Estimates of multifactor random-effects regression models.

| Variable Name | Fixed-Effects Regression Results | | | Random-Effects Regression Results | |
|---|---|---|---|---|---|
| | Regression Coefficient | Standard Error | *t*-Test | Variance Component | Chi-Squared Test |
| $\gamma_{00}$ | 13.2053 | 0.8196 | 16.112 *** | 222.4776 | 141.8825 *** |
| HEAL | −1.9220 | 0.4144 | −4.638 *** | 3.3173 | 45.7651 |
| RJGDMJ | −0.1843 | 0.0460 | −4.006 *** | 0.0344 | 47.4904 |
| ZRGDB | −0.0374 | 0.0087 | −4.300 *** | 0.0022 | 55.0921 * |
| SCGJ | −2.3390 | 0.5426 | −4.310 *** | 21.1288 | 59.3682 ** |
| NYSRB | −0.0792 | 0.0075 | −10.522 *** | 0.0124 | 84.2431 *** |
| FNSRB | 0.0003 | 0.0078 | 0.036 | 0.0108 | 57.9672 * |
| CGB | −3.8157 | 1.2928 | −2.951 *** | 22.7815 | 38.7926 |

Note: $p < 0.01$, extremely significant, labeled ***; $p < 0.05$, significant, labeled **; $p < 0.1$, generally significant, labeled *.

Comparing the level-1 random-effects estimation results of the multifactor random-effects regression model and the null model [55], this study calculated the overall explanatory power of the level-1 explanatory variables on the explained variables and found that the variance shrinkage ratio of the level-1 random term was $(521.0150 − 484.4892)/521.0150 = 0.0701$, which indicated that the head of the household being relatively healthy, per capita area of cropland, ratio of transferred cropland, ownership of large agricultural production machinery or livestock for agricultural production, ratio of agricultural income, ratio of non-farm income,

and presence of village cadres can explain 7.01% of the within-group variance in the rate of abandonment of farming by farm households.

*4.4. Intercept Model Analysis*

The intercept term in the level-1 model was used as an explained variable, and the level-1 variables were added to the level-1 model in a non-centered manner to construct an intercept model in order to test whether the village-level variables have a significant effect on the abandonment rate. The specific form of the model is shown below:

Level-1 model:

$$QGZB_{ij} = \beta_{0j} + \gamma_{ij} \tag{5}$$

Level-2 model:

$$\beta_{0j} = \gamma_{00} + \gamma_{01}TQJL_j + \gamma_{02}DLWZ_j + \gamma_{03}CZDX_j + \gamma_{04}CZNYSCB_j + \gamma_{05}FNQY_j \\ + \gamma_{06}HNFW_j + \gamma_{07}TDZY_j + \mu_{0j}$$

In Equation (5), $TQJL_j$ is the commuting distance to the *j*th village, and the rest of the level-2 explanatory variables have similar meanings, which represent the physical geography and socio-economic situation of each village; the meanings of all other variables are the same as in Equation (1).

The fixed-effects estimation results show that the abandonment rates are all affected to varying degrees by the physical geographic and socio-economic characteristics of the villages.

Table 7 shows that the regression coefficients for the intercept term $\beta_0$ for the five variables of commuting distance, namely whether it is a suburb of a large or medium-sized city, if the village topography is plain, the ratio of the number of people in agricultural production in the village, and whether it has experienced land expropriation passed the test of significance. Among them, commuting distance, whether the topography of the village is plain or not, and the ratio of the number of people in agricultural production in the village have extremely significant negative effects on the rate of abandonment of farming; whether it is a suburb of a large or medium-sized city has a generally significant negative effect on the rate of abandonment of farming, in the opposite direction of the expected effect; and whether it is experiencing land expropriation has a significant positive effect on the rate of abandonment of farming. Increased commuting distance reduces the rate of abandonment of farming, and in villages with long commuting distances, farmers in the village have a stronger "love of the land" and a lower degree of convenience in engaging in non-agricultural employment, which makes them more inclined to engage in agricultural production activities [67,68]. The flatter the topography of the village, the lower the abandonment rate, and flatter topography is more conducive to the scale and mechanization of farmland management, to improving agricultural income, and to maintaining the motivation of farmers in agricultural production [11,69]. The higher the ratio of the number of people engaged in agricultural production in the village, the lower the abandonment rate because farmers are prone to herd behavior in such a production environment and continue to operate the farmland; at the same time, a large number of farmers engaged in agricultural production in the village also provide a relatively broad market for the transfer of agricultural land, which also reduces the rate of abandonment of the farmland [15,70]. The village is on the outskirts of a large or medium-sized city, which reduces the rate of abandonment of cropland, and its proximity to a large or medium-sized city provides, on the one hand, a relatively wide market for the agricultural activities of farmers and facilitates the sale of agricultural products, and on the other hand, the convenient location provides the possibility for farmers to engage in both agricultural and non-agricultural production activities at the same time. In villages that have experienced land expropriation, the abandonment rate will increase because land expropriation will make part of the farmers' farmland present fragmentation characteristics, resulting in a

reduction in the area of farmland available for production and operation, which is not conducive to the realization of large-scale management and the transfer of farmland and trusteeship, reducing the enthusiasm of farmers for agricultural production [15,69] and prompting an increase in the rate of abandonment of farmland.

**Table 7.** Intercept model fixed-effects estimation results.

| Variable Name | Regression Coefficient | Standard Error | *t*-Test |
|---|---|---|---|
| $\gamma_{00}$ | 15.3173 | 1.7085 | 8.965 *** |
| TQJL | −0.2571 | 0.0682 | −3.771 *** |
| DLWZ | −2.5563 | 1.3533 | −1.889 * |
| CZDX | −4.7083 | 0.8881 | −5.302 *** |
| CZNYSCB | −0.0586 | 0.0147 | −3.993 *** |
| FNQY | 0.0227 | 1.0449 | 0.022 |
| HNFW | 0.2797 | 1.3159 | 0.213 |
| TDZY | 1.9319 | 0.9553 | 2.022 ** |

Note: $p < 0.01$, extremely significant, labeled ***; $p < 0.05$, significant, labeled **; $p < 0.1$, generally significant, labeled *.

Comparing the Level-2 random-effect estimates of the intercept model and the null model, the variance shrinkage ratio of the Level-2 random term was calculated to be 0.0956, indicating that the village-level variables could explain 9.56% of the between-group variation in abandonment rate. The within-group conditional correlation coefficient was further calculated to be 15.04%, indicating that the proportion of inter-village variance to the total variance was reduced with the inclusion of the five village variables mentioned above.

*4.5. Full Model Analysis*

The full model was constructed based on the results of the multifactor random-effects regression model and intercept model analysis. Variables with significant effects on abandonment rate at the farm household level were included in the Level-1 model, in which continuous variables for the moderating effect test were added with group mean centering. Variables with significant effects of the village level on abandonment rate were included in the Level-1 model with the intercept as an explained variable; at the same time, these significant variables were included in the Level-1 model with the slope of farm-level variables with significant group differences as the explained variable, to which continuous-type variables for the test of moderating effect were added by centering the total mean to verify the effect of village level factors on the effect of farm-level factors on the moderating effect of the effect of abandonment rate. The specific form of the model is shown below:

Level-1 model:

$$QGZB_{ij} = \beta_{0j} + \beta_{1j}HEAL_{ij} + \beta_{2j}RJGDMJ_{ij} + \beta_{3j}\left(ZRGDB_{ij} - \overline{ZRGDB}_{.j}\right) + \beta_{4j}SCGJ_{ij}$$
$$+ \beta_{5j}\left(NYSRB_{ij} - \overline{NYSRB}_{.j}\right) + \beta_{6j}CGB_{ij} + \gamma_{ij} \tag{6}$$

Level-2 model:

$$\beta_{0j} = \gamma_{00} + \gamma_{01}TQJL_j + \gamma_{02}DLWZ_j + \gamma_{03}CZDX_j + \gamma_{04}CZNYSCB_j + \gamma_{05}TDZY_j + \mu_{0j}$$
$$\beta_{1j} = \gamma_{10} + \mu_{1j}$$
$$\beta_{2j} = \gamma_{20} + \mu_{2j}$$
$$\beta_{3j} = \gamma_{30} + \gamma_{31}\left(TQJL_j - \overline{TQJL}_.\right) + \gamma_{32}DLWZ_j + \gamma_{33}CZDX_j$$
$$+ \gamma_{34}\left(CZNYSCB_j - \overline{CZNYSCB}_.\right) + \gamma_{35}TDZY_j + \mu_{3j}$$
$$\beta_{4j} = \gamma_{40} + \gamma_{41}\left(TQJL_j - \overline{TQJL}_.\right) + \gamma_{42}DLWZ_j + \gamma_{43}CZDX_j$$
$$+ \gamma_{44}\left(CZNYSCB_j - \overline{CZNYSCB}_.\right) + \gamma_{45}TDZY_j + \mu_{4j}$$
$$\beta_{5j} = \gamma_{50} + \gamma_{51}\left(TQJL_j - \overline{TQJL}_.\right) + \gamma_{52}DLWZ_j + \gamma_{53}CZDX_j$$
$$+ \gamma_{54}\left(CZNYSCB_j - \overline{CZNYSCB}_.\right) + \gamma_{55}TDZY_j + \mu_{5j}$$

$$\beta_{6j} = \gamma_{60} + \mu_{6j}$$

In Equation (6), the variables have the same meaning as in Equations (1), (4), and (5).

As can be seen from Table 8, the effects of significant variables at the farm household level and village level on the abandonment rate are similar to those estimated by the multifactor random-effects regression model, as well as the intercept model. In terms of village-level variables, commuting distance, whether the village terrain is plain or not, and the ratio of the number of people in agricultural production in the village, they still have extremely significant negative effects on the abandonment rate of farm households; whether it is a suburb of a large or medium-sized city still has a generally significant negative effect on the abandonment rate of farm households; and whether one has experienced land expropriation or not has a significant positive effect on the abandonment rate of farm households, with only slight differences in the size of the regression coefficients, which may be due to the introduction of village-level significant variables in the full model. A possible reason for this is that there is a certain moderating effect of the village-level variables introduced by the full model [57]. Based on the estimation results of the full model, this study focuses on analyzing the moderating effect of village-level variables on the effect of farm-household-level variables on abandonment rate. When the regression coefficients of the village-level variables have the same sign as the regression coefficients of the farm-household-level variables, it means that the village-level variables strengthen the influence of the farm-household-level variables on the abandonment rate; when the regression coefficients of the village-level variables have the opposite sign of the regression coefficients of the farm-household-level variables, it means that the village-level variables weaken the influence of the farm-household-level variables on the abandonment rate [56].

The significant moderating effects of the estimation results are as follows: (1) the slope of the commuting distance and the ratio of transferred farmland are generally significantly positively correlated with a regression coefficient of 0.0028, and the ratio of transferred farmland is extremely significantly negatively correlated with the abandonment rate, so the further the commuting distance is from the village, the weaker the negative correlation relationship will be between the ratio of transferred farmland and the abandonment rate. The slope of the ratio of the number of people in agricultural production in villages and the ratio of transferred farmland is generally significantly positively correlated, with a regression coefficient of 0.0007, whereas the ratio of transferred farmland is extremely significantly negatively correlated with the rate of abandoning farmland; therefore, the higher the ratio of the number of people in agricultural production in villages is, the weaker the negative correlation between the ratio of transferred farmland and the rate of abandoning farmland is; (2) The slope of the relationship between whether the village is a suburb of a large or medium-sized city and the ownership of large agricultural production machinery or livestock for agricultural production is generally significantly negatively correlated, with a regression coefficient of −2.3175; thus, the villages in the suburbs of a large or medium-sized city are extremely significantly negatively correlated with the rate of abandonment of cropland. The negative correlation between the availability of large agricultural production equipment or livestock and the abandonment rate is stronger in villages that are suburbs of large and medium-sized cities. There is a significant positive correlation between the topography of the village and the slope of the ownership of large agricultural production machinery or livestock for agricultural production, with a regression coefficient of 3.0912, whereas there is a highly significant negative correlation with the abandonment rate of the village; therefore, if the topography of the village is plain, the negative correlation between the slope of the ownership of large agricultural production machinery or livestock for agricultural production and the abandonment rate of the village is weaker. The negative correlation between the ownership of large agricultural production machinery or livestock and the abandonment rate is weaker; (3) The slope of commuting distance and agricultural income ratio is generally significantly positively correlated, with a regression coefficient of 0.0020, and the agricultural income ratio is extremely significantly negatively correlated

with the abandonment rate, so the further the commuting distance, the weaker the negative correlation of agricultural income ratio and abandonment rate. The slope of the agricultural income ratio is significantly positively correlated with village topography, with a regression coefficient of 0.0251 and a highly significant negative correlation with abandonment rate, so the weaker the negative correlation between the agricultural income ratio and abandonment rate, the more the village topography is plain. There is a highly significant negative correlation with the slope of the agricultural income ratio, with a regression coefficient of $-0.0364$, and a highly significant negative correlation with the abandonment rate, so that the villages that have experienced land acquisition have a stronger negative correlation between the agricultural income ratio and the abandonment rate.

**Table 8.** Full model fixed-effects estimation results.

|  | Variable Name | Regression Coefficient | Standard Error | *t*-Test | *p*-Value |
|---|---|---|---|---|---|
|  | Intercept term$\gamma_{00}$ | 12.0579 | 0.7842 | 15.377 | 0.000 |
|  | TQJL$\gamma_{01}$ | $-0.2528$ | 0.0677 | $-3.735$ | 0.000 |
| $\beta_0$ | DLWZ$\gamma_{02}$ | $-2.4540$ | 1.3157 | $-1.865$ | 0.062 |
|  | CZDX$\gamma_{03}$ | $-4.7223$ | 0.8827 | $-5.350$ | 0.000 |
|  | CZNYSCB$\gamma_{04}$ | $-0.0548$ | 0.0148 | $-3.690$ | 0.000 |
|  | TDZY$\gamma_{05}$ | 1.8234 | 0.9293 | 1.962 | 0.050 |
| $\beta_1$ | HEAL$\gamma_{10}$ | $-1.9073$ | 0.4164 | $-4.581$ | 0.000 |
| $\beta_2$ | RJGDMJ$\gamma_{20}$ | $-0.1975$ | 0.0503 | $-3.927$ | 0.000 |
|  | ZRGD $\gamma_{30}$ | $-0.0541$ | 0.0125 | $-4.338$ | 0.000 |
|  | ZRGDB$\times$TQJL$\gamma_{31}$ | 0.0028 | 0.0015 | 1.820 | 0.069 |
| $\beta_3$ | ZRGDB$\times$DLWZ$\gamma_{32}$ | $-0.0387$ | 0.0290 | $-1.337$ | 0.182 |
|  | ZRGDB$\times$CZDX$\gamma_{33}$ | 0.0058 | 0.0164 | 0.354 | 0.723 |
|  | ZRGDB$\times$CZNYSCB$\gamma_{34}$ | 0.0007 | 0.0003 | 2.440 | 0.015 |
|  | ZRGDB$\times$TDZY$\gamma_{35}$ | 0.0111 | 0.0188 | 0.592 | 0.554 |
|  | SCGJ$\gamma_{40}$ | $-3.5314$ | 0.8546 | $-4.132$ | 0.000 |
|  | SCGJ$\times$TQJL $\gamma_{41}$ | 0.0038 | 0.0879 | 0.044 | 0.966 |
| $\beta_4$ | SCGJ$\times$DLWZ$\gamma_{42}$ | $-2.3175$ | 1.3835 | $-1.675$ | 0.094 |
|  | SCGJ$\times$CZDX$\gamma_{43}$ | 3.0912 | 1.2162 | 2.542 | 0.012 |
|  | SCGJ$\times$CZNYSCB$\gamma_{44}$ | 0.0071 | 0.0201 | 0.352 | 0.725 |
|  | SCGJ$\times$TDZY$\gamma_{45}$ | $-1.1113$ | 1.2374 | $-0.898$ | 0.370 |
|  | NYSRB$\gamma_{50}$ | $-0.0685$ | 0.0097 | $-7.093$ | 0.000 |
|  | NYSRB$\times$TQJL$\gamma_{51}$ | 0.0020 | 0.0011 | 1.849 | 0.064 |
| $\beta_5$ | NYSRB$\times$DLWZ$\gamma_{52}$ | 0.0026 | 0.0224 | 0.116 | 0.908 |
|  | NYSRB$\times$CZDX$\gamma_{53}$ | 0.0251 | 0.0120 | 2.092 | 0.037 |
|  | NYSRB$\times$CZNYSCB$\gamma_{54}$ | $-0.0000$ | 0.0002 | $-0.018$ | 0.985 |
|  | NYSRB$\times$TDZY$\gamma_{55}$ | $-0.0364$ | 0.0127 | $-2.862$ | 0.005 |
| $\beta_6$ | CGB$\gamma_{60}$ | $-3.6382$ | 1.2818 | $-2.838$ | 0.005 |

From Table 9, it can be seen that after the introduction of the five level-2 variables of commuting distance, namely whether it is a suburb of a large or medium-sized city, whether the village is a plain, the ratio of the number of people in agricultural production in the village, and whether it has experienced land expropriation, the random effects of the ratio of the cropland that was transferred to cropland, the ratio of the number of people in agricultural production that owns a large-scale agricultural production machine or livestock used for agricultural production, and the ratio of the agricultural income are still significant, which indicates that the introduction of level-2 variables to the level-2 variables needs to be enhanced, and further village-level-related variables or higher-level variables need to be added for analysis [55]. Meanwhile, after further comparing the random-effect estimation results of the full model random-effect regression results and the multifactor random-effect regression model, as well as calculating the variance reduction ratio, it was found that $(222.4776 - 130.3265)/222.4776 = 0.4142$; that is to say that 41.42% of the difference in the mean value of abandonment of farmland rate in villages could be explained by the above five level-2 variables.

**Table 9.** Full model random-effects regression results.

| Variable Name | Random-Effects Regression Results | |
|---|---|---|
| | Variance Component | Chi-Squared Test |
| $\gamma_{00}$ | 130.3265 | 116.5629 *** |
| HEAL | 2.7051 | 49.5024 |
| RJGDMJ | 0.0773 | 43.2623 |
| ZRGDB | 0.0027 | 52.8242 * |
| SCGJ | 35.0686 | 59.8170 ** |
| LNRJSR | 8.7962 | 98.1779 |
| NYSRB | 0.0080 | 86.9579 *** |
| CGB | 19.6672 | 34.9425 |

Note: $p < 0.01$, extremely significant, labeled ***; $p < 0.05$, significant, labeled **; $p < 0.1$, generally significant, labeled *.

## 5. Conclusions

(1) A total of 83.63% of the differences in the abandonment rate of farm households are caused by the differences in farm households, and the farm-household-level factor is the main cause; 16.37% is caused by the differences in their villages, and the village-level factor should not be underestimated. In terms of background effects, village-level factors not only have a direct effect on the average level of abandonment rate in each village, but also a moderating effect on the effect of farm-household-level factors on abandonment rate;

(2) The fixed-effects estimates of the single-factor and multifactor random-effects regression models show that the differences in farm abandonment rates are affected by all of the human, natural, physical, financial, and social capitals of farm households. Based on the fixed-effects estimates of the multifactor random-effects regression model, whether the head of the household is healthier, the per capita area of cropland, the ratio of transferred cropland, the possession of large-scale agricultural production machinery for agricultural production or livestock, agricultural income ratio, and whether or not they have village cadres all have a significant negative effect on the abandonment rate. Meanwhile, the random-effect estimation results of the multifactor random-effect regression model show that there are significant village differences in the mean value of abandonment rate of farm households in each village, and there are also significant between-group differences in the negative impacts of the ratio of transferred cropland, the possession of large-scale agricultural production machinery or livestock used in agricultural production, and the ratio of agricultural income on the abandonment rate;

(3) The fixed-effects estimation results of the intercept model showed that commuting distance, whether it is a suburb of a large or medium-sized city, the topography of the village is plain or not, and the ratio of the number of people in agricultural production in the village have significant negative effects on the abandonment rate, while the experience of land expropriation or not has a significant positive effect on the abandonment rate;

(4) The results of the full model show that the farm-household-level variables with significant fixed effects in the multifactor random-effects regression model, as well as the village level variables with significant fixed effects in the intercept model, can still pass the significance test and the direction of the effect remains the same. In terms of moderating effects, the slopes of commuting distance and the ratio of the number of people in agricultural production in villages are significantly positively correlated with the ratio of transferred cropland, weakening the negative correlation between the ratio of transferred cropland and the rate of abandonment of cropland. The slope of whether the village is a suburb of a large or medium-sized city is significantly negatively correlated with the ownership of large agricultural production implements or livestock, which strengthens the negative correlation between the ownership of

large agricultural production implements or livestock and the abandonment rate; whereas the slope of whether the village is a plain is significantly positively correlated with the ownership of large agricultural production implements or livestock and weakens the negative correlation between the ownership of large agricultural production implements or livestock and the abandonment rate. Commuting distance and whether the village topography is plain are significantly positively associated with the slope of the agricultural income ratio, weakening the negative association between the agricultural income ratio and abandonment, and whether the village has experienced land confiscation is significantly negatively associated with the agricultural income ratio, strengthening the negative association between the agricultural income ratio and abandonment;

(5)  By calculating the variance reduction ratios of the random-effects regression model, the intercept model, and the complete model, it can be found that the factors at the household level can to some extent effectively explain the intra-group differences in in the levels of cropland abandonment. The factors at the village level can, to some extent, effectively explain the inter-group differences in the levels of cropland abandonment and the differences in the mean values of abandonment rates of the villages.

**Author Contributions:** Conceptualization, X.W.; Methodology, X.W.; Software, D.Z.; Data curation, D.Z.; Writing—original draft, D.Z.; Supervision, X.W. All authors have read and agreed to the published version of the manuscript.

**Funding:** This research received no external funding.

**Data Availability Statement:** Data for the study can be obtained from the publicly available China Labor Force Dynamics Survey database.

**Conflicts of Interest:** The authors declare no conflict of interest.

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
