# Peer review of "Study on the Causes of Differences in Cropland Abandonment Levels among Farming Households Based on Hierarchical Linear Model—13,120 Farming Households in 26 Provinces of China as an Example"

_land, doi:10.3390/land12091791_

Round 1

Reviewer 1 Report

The article is devoted to finding the reasons that lead to the abandonment of agricultural land. The study was carried out for the territory of China. The task of the work is to simulate the influence of various factors on the decision to abandon the land. This problem is solved using hierarchical linear regression. The strength of the work is the use of modern methods of statistical analysis and a large amount of analyzed empirical material.

The weak side of the work is that the authors, when using statistical analysis, did not resort to the use of statistical graphics. In addition, the debatable point is the detail of the description of the theoretical foundations of the work. The theoretical foundations are described in excessive detail. For this purpose, a separate section has been created in the structure of the article (section 2). This material can be submitted in short form by combining sections 1 and 2. On the other hand, this material is presented in a very interesting way. From a reader's point of view, it deserves to be kept as it is.

The literary review is done at a qualitative level. A fairly large number of literary sources were used. It should be noted that the authors used not only the literature on research in China. They also used works describing patterns of abandonment of agricultural land in other regions of the world. The bibliography is new and up-to-date. Most of the works included in it have appeared in the last five years.

In the review, special attention is paid to listing the methods of regression analysis used to model the abandonment of agricultural land. Also, based on the analysis of literary sources, a classification of factors influencing the abandonment of agricultural land is given. These are very valuable generalizations made by the authors based on a review of publications.

Data analysis methods are described in sufficient detail and clearly. The preliminary data processing and their statistical analysis are described separately. The authors do not just mention the use of a hierarchical linear model. They elaborate on the essence of this regression analysis method. The description is accompanied by formulas. Such attention to methodological detail is essential. Hierarchical linear regression is currently not yet a ubiquitous method of analysis. This work may contribute to the popularization of this method.

The results are described quite clearly. The conclusions obtained are consistent with the results. The most interesting is the relationship between the role of differences between households and differences between villages, described based on the results of the work. The authors found that the share of abandoned agricultural land is more influenced by the characteristics of households than by the characteristics of villages. This is a valuable observation that would be interesting to test in other regions. Perhaps this article will be able to draw the attention of other scientists to this topic.

I believe that the work has scientific value, is interesting for readers and can be published in its current form. But if the authors can add an illustration to the article (optional), it will make it even more interesting for readers.

Author Response

In conjunction with the review comments, the authors have completed revisions to the content of the paper, please see attached.

Reviewer 2 Report

1. Whether the content of hierarchical linear modeling in the introduction part can be placed in the third part for detailed explanation;

2. The theoretical analysis part can be integrated and simplified by further integrating the "rational man" assumption with the analytical content of the sustainable livelihood analysis framework; at the same time, the theoretical analysis framework can be changed into an explanatory framework to further clarify the characteristics of stratification;

3. In the discussion and conclusion section, the discussion should be written first and then the conclusion, and the discussion section can be divided into three parts, including: the contribution of the paper, the policy recommendations, and the shortcomings of the study.

Extensive editing of English language required

Author Response

(The authors gave the same response as above.)

Reviewer 3 Report

First of all, I would like to commend authors for the very well elaborated paper and considering study of this issue that is critical to the contemporary development and important. for the Land journal. 

Line 47  / Suggestion to authors to refer to the SDGs. This could help authors as well to clearly indicate the purpose and goals of the study. 

Line 49 – Line 48 / The introductory section should be structured as overview of the state of the art - authors are advised to introduce subsection and to clearly indicate levels of the analysis at the beginning of the paragraph. There are too many lists and numberings (aspects, factors, categories, and it is very hard to follow. The table would be beneficial as well. It is very hard to follow the paper structure – Authors should provide a concluding paragraph of the introduction section to introudce the structure. Grpahical abstract would be beneficial as well. 

Line 85 – 119 / Paper would benefit of summarizing the results for cropland abandonment in relation to the existing theoretical research. Also, authors should interelate their results with these theoretical standpoints and try to avoid use of “this may be because” and istead use – “this confirms the viewpoint/or these follows the line of reasoning of ….”

Line 210 Figure 2 / I would recommend authors to clearly and graffically differentiate what comes from the rational man hypothesis and what comes from the sustainable livelihood analyses. Please review the frame with External shcoks – the part of text is missing. 

Line 236 – Line 240 – Hypothesis are very broad and not easy to understand – which five types of livelihood capital , which characterizing factors, what significant effects. In summary, hypothesis have two many uknown elements. 

Methodology section – This section is well elaborated. I would appreciate to see the geographical distribution of the municipalities covered in this research – as a figure. 

Analysis section – Even though the analysis seems very deep and detaild, these is set and only briefly explained – I strongly discourage using “This may be bauces” This part should be foucsed on analysis and all methodological information (regarding statistical methods and models) should be moved to methodological section. 

Line675 / It is not logical to have discussion after conlussion. Conclussion just summarizes the analysis section and do not follow the hypothesis set in the introduction. Authors should reply to their hypothesis (or reformultate them). Additionally, I would encourage authors to try to think about the usufulness of the results and the domain of application. 

Disscussion – Authors did not managed to provide any disscussion. n this section, I sicerely think authors should try to provide additional explations on the causes and differences in the researched sample – as stated in the title. 

After reading 30 pages long paper and seeing this sentence> “Due to the limitation of time and effort, the data used in this study are second-hand data” I was very disapointed. With the assumption that there is language barrier, authors are advised to use terminology of secondary/primary data, explain the duration and costs of the study, etc. I decline to believe that official state statistic is not sufficient for the quality study. At contrary, I believe that lot of effort is put in this study and it deserves to be clearly explained and ellaborated. Limitations should be clearly ellaborated but in relation to hypothesis. Afterwards, authors should express future steps in their study. 

I have checked the box with moderate english editing since the analysis section is well written but introduction section and theoretical background should be majorly improved to be understandable. 

Author Response

(The authors gave the same response as above.)
